Effect of exercise based interventions on sleep and circadian rhythm in cancer survivors—a systematic review and meta-analysis

Gururaj Rachita 1
Samuel Stephen Rajan stephen.samuel@manipal.edu 2 3
Kumar K Vijaya 2
Nagaraja Ravishankar 4
Keogh Justin W.L. 2 5 6
1 Ramaiah College of Physiotherapy, Ramaiah University of Applied Sciences , Bengaluru , Karnataka , India
2 Physiotherapy, Kasturba Medical College, Mangalore, Manipal Academy of Higher Education , Manipal , Karnataka , India
3 Cancer Control Division, Department of Surgery University of Rochester Medical Center, University of Rochester , Rochester , NY , United States of America
4 Department of Biostatistics, Vallabhbhai Patel Chest Institute, University of Delhi , New Delhi , New Delhi , India
5 Faculty of Health Sciences and Medicine, Bond University , Gold Coast, Australia , Australia
6 Human Potential Centre, AUT University , Auckland , New Zealand
Neiva Henrique
Electronic publication date: 2024 Mar 8
Publication date: 2024
Volume: 12
Electronic Location ID: e17053
Received 2023 Oct 6; Accepted 2024 Feb 13
Copyright: ©2024 Gururaj et al.
Copyright year: 2024
Copyright holder: Gururaj et al.
License: This is an open access article distributed under the terms of the Creative Commons Attribution License, which permits unrestricted use, distribution, reproduction and adaptation in any medium and for any purpose provided that it is properly attributed. For attribution, the original author(s), title, publication source (PeerJ) and either DOI or URL of the article must be cited.
License URL: https://creativecommons.org/licenses/by/4.0/

Keywords: Carcinoma, Circadian Rhythm, Exercise, Sleep disturbance, Tai chi, Yoga, Endurance training, Strength training

Funding: The authors received no funding for this work.

==============================
Background

Disrupted circadian rhythm commonly reported in cancer survivors is closely associated with cancer related fatigue, sleep disturbances and compromised quality of life. As more cancer survivors request non-pharmacological treatment strategies for the management of their chronic sleep-related symptoms, there is a need for meta-analyses of various interventions such as exercise on sleep and circadian rhythm disturbances.

Methods

A search for RCT’s was conducted in April 2020 and updated in July 2023 using relevant keywords for cancer, sleep, circadian rhythm and exercise interventions on PubMed, Scopus, Web of Science, PEDro and CINAHL.

Results

Thirty-six studies were included for qualitative analysis and 26, for meta-analysis. Thirty-five studies analyzed sleep outcomes, while five analyzed circadian rhythm. RCT’s studying the effect of aerobic exercise, resistance exercise, combined aerobic and resistance exercise, physical activity, yoga, or tai chi were included. Meta-analysis results showed significant exercise-related improvements on sleep quality assessed by Pittsburgh Sleep Quality index (PSQI) (SMD = −0.50 [−0.87, −0.13], p = 0.008), wake after sleep onset (WASO) (SMD = −0.29 [−0.53, −0.05], p = 0.02) and circadian rhythm, assessed by salivary cortisol levels (MD = −0.09 (95% CI [−0.13 to −0.06]) mg/dL, p < 0.001). Results of the meta-analysis indicated that exercise had no significant effect on sleep efficiency, sleep onset latency, total sleep time and circadian rhythm assessed by accelerometry values.

Conclusion

While some sleep and circadian rhythm outcomes (PSQI, WASO and salivary cortisol) exhibited significant improvements, it is still somewhat unclear what exercise prescriptions would optimize different sleep and circadian rhythm outcomes across a variety of groups of cancer survivors.

Implication

As exercise does not exacerbate cancer-related circadian rhythm and sleep disturbances, and may actually produce some significant benefits, this meta-analysis provides further evidence for cancer survivors to perform regular exercise.

Introduction

Circadian rhythm or circadian clock refers to the approximately 24-hour cycle of the body at the physiological, molecular and psychological level (Fu & Kettner, 2013) which regulates a variety of biological processes such as sleep-wake cycle, body temperature, homeostasis of blood glucose, hormone secretion and physical activity (Serin & Aca. Tek, 2019). Cancer survivors are more likely to present with disrupted circadian rhythm in comparison to general population not only during- or post-treatment but even prior to the initiation of the primary cancer treatment (Miaskowski et al., 2011). This de-regulated circadian rhythm in cancer survivors is associated with a poor quality of life and prognosis, high mortality and sleep disturbances (Miaskowski et al., 2011; Innominato et al., 2009; Strøm et al., 2022).

Sleep disturbances refer to “disorders of initiating and maintaining sleep (DIMS, insomnias), disorders of excessive somnolence (DOES), disorders of sleep–wake schedule, and dysfunctions associated with sleep, sleep stages, or partial arousals (parasomnias)” (Cormier, 1990). Sleep disturbance in cancer survivors is a chronic disabling problem with a high prevalence rate ranging from 30% to 50% (Miaskowski et al., 2011). The underlying mechanisms contributing to these disturbances are believed to be multifactorial, including a host of physical and psychological components (Balachandran et al., 2021). These circadian rhythm and sleep disturbances are also thought to affect and be affected by a host of other physiological functions, with dysregulation of clock genes and bidirectional interplay between driver oncogenes on the clock genes common in cancer survivors (Balachandran et al., 2021). The multifactorial nature of these disturbances in cancer survivors suggests that a single therapeutic approach (with pharmacological approaches typically used initially), may not adequately address all the mechanisms contributing to these issues.

While common pharmacological treatments prescribed for sleep disorders can significantly improve sleep outcomes in cancer survivors, they also have numerous side-effects (Fitzgerald & Vietri, 2015). As a result, several reviews have now been completed on non-pharmacological treatments including cognitive behavioral therapy (CBT) (Fitzgerald & Vietri, 2015; Garl et al., 2014; Melton, 2018; Johnson et al., 2016) and relaxation therapy and therapeutic massage (Samuel et al., 2020) on whether they can improve sleep disturbances for cancer survivors. Exercise may also be a potentially useful non-pharmacological intervention as it has a variety of beneficial effects on physical, psychological and psychosocial outcomes, with relatively minimal side-effects if progressed gradually (Carayol et al., 2013; Samuel et al., 2019; Samuel et al., 2013; Kokila & Smitha, 2017). Some preliminary evidence also suggests that exercise may significantly benefit circadian rhythm deregulation and sleep disturbances in cancer survivors in some studies; however, previous meta analyses have shown no significant overall benefit on sleep quality (Bernard et al., 2019; Mercier, Savard & Bernard, 2017; Yang et al., 2021).

Exercise is defined as “a subset of physical activity, requiring specifically planned, structured, and repetitive movements with a goal of improving performance or fitness” (Caspersen, Powell & Christenson, 1985). Modes of exercise that have been examined in cancer survivors include general physical activities such as walking, as well as more structured exercise modes including aerobic exercise, resistance exercise, yoga and tai chi. Although there is some evidence suggesting that exercise may potentially improve certain aspects of sleep in cancer survivors (Bernard et al., 2019; Mercier, Savard & Bernard, 2017), these reviews still have their limitations. There is lacuna of evidence that encapsulates all forms of exercise, with most studies involving resistance or aerobic exercise common in the Western World compared to other forms of exercise such as yoga and tai chi that originated in Asian countries. There is also lack of evidence summarizing the effect of exercise on circadian rhythm in cancer survivors. Further, previous reviews either had no restriction on the study design and included studies other than RCTs (Mercier, Savard & Bernard, 2017) or excluded RCTs which had no individual patient data (Bernard et al., 2019). Hence, this systematic review was developed with an objective to summarize RCT level evidence of all forms of exercise (including aerobic training, resistance training, combined training, physical activity, yoga and tai chi) on sleep and circadian rhythm outcomes across all groups of cancer survivors.

Materials & Methods

Registration

The review protocol was registered on Open Science Framework on 24th of March 2020 and can be accessed at https://osf.io/aktdn.

Search strategy

Studies were identified by running a search on PubMed, Scopus, Web of Science, CINAHL and PEDro from inception to 1st of May 2020. The search was re-run on 27th of July 2023 to include eligible studies published in the interim. The search strategy included the following terms for cancer: Carcinoma [MeSH], Cancer, Neoplasms [MeSH], Malignancy, exercise- Physiotherapy, Physical therapy, Exercise [MeSH], Yoga, Tai chi, Exercise therapy [MeSH], sleep- Sleep disturbances, Insomnia, Sleep wake disorders [MeSH], Sleep initiation and maintenance disorders [MeSH], sleep disorders and circadian rhythm [MeSH]. The search terms were combined using Boolean operators ‘AND’ or ‘OR’ wherever relevant. For example the search strategy used in PubMed was as follows: (((((((((((carcinoma[MeSH]) OR cancer) OR neoplasms[MeSH]) OR malignancy)) AND (((((sleep disturbances) OR insomnia) OR sleep wake disorders[MeSH]) OR (sleep initiation and maintenance disorders[MeSH])) OR sleep disorders)) AND (((((((((physiotherapy) OR physical therapy) OR exercise[MeSH]) OR yoga) OR tai chi) OR relaxation techniques) OR physical therapy modalities[MeSH]) OR electrotherapy) OR exercise therapy[MeSH]. A more detailed description of the full search strategy is provided in a Supplementary File.

In addition, the references of each included study were screened for other potentially eligible articles.

Selection of studies

The articles were checked for inclusion based on the pre-set inclusion criteria- (1) including participants with any form or stage of cancer, (2) including any form of exercise including general physical activity, yoga and tai chi, (3) measuring sleep disturbances and/or circadian rhythm as their outcome, (4) published in English language and (5) involving a randomized controlled trial (RCTs) or Quasi-RCT design (where allocation is not truly random). After the results of each database were exported to a reference management software and duplicates removed, screening of title and abstract were performed independently by two reviewers (SRS and RG). The same two reviewers performed the full text screening of the eligible studies with reasons mentioned for exclusion. Discrepancies between the reviewers regarding selection of any study was sorted by discussion between the two reviewers (SRS and RG) and two other researchers (VK and JK).

Extraction of data and management

Relevant data from the included studies was extracted using a template developed on Microsoft Excel which included descriptors of the study title, author, year of publication, journal, aims, study population, sample size, study design, method of randomization, allocation, adjuvant therapy, study duration, intervention details including type and duration with the exercise prescription, details of the comparators, outcome measures, type of data (of the outcome measure), results and adverse events. The data extraction was done based on the exercise type and the outcome measures used by the included studies for the evaluation of changes in sleep and circadian rhythm was clearly stated during extraction. Post intervention data was considered for extraction. For studies that did not report immediate post data, the values mentioned in the first follow up was extracted. In case of missing data, the authors were contacted for retrieval of the data. Double extraction of data was performed by SRS and RG and checked by VK for clarification in case of disagreements.

Outcome variables

The outcome variables extracted from the eligible studies included objective and subjective sleep and circadian rhythm outcomes.

Assessment of risk of bias

The included studies were assessed for risk of bias using the Cochrane Risk of Bias Tool (Higgins et al., 2011) independently by two reviewers, SRS and RG. Any discrepancies between the reviewers were resolved by discussion between the two reviewers (SRS and RG) and JK. The studies were assessed under the following domains- random sequence generation, allocation concealment, blinding of participants and personnel, blinding of outcome assessment, incomplete outcome data, selective reporting, and other bias. The studies were rated as high risk if the answer was a ‘no’ for the first three domains and ‘yes’ for the latter three domains. The studies were rated as having a low risk of bias if the answer was a ‘yes’ for the initial three domains and ‘no’ for the latter three domains. An unclear risk was marked if there was missing data or unknown risk. If found, the protocols of the included studies were checked for missing data and certainty of an outcome.

Statistical analysis

Meta-analysis was performed for the studies that provided adequate data specific to predefined outcomes or else, qualitative analysis was performed to such studies. Studies that included usual care as comparison were included for the meta-analysis, others were qualitatively summarised. Standardized mean difference was typically used as a measure of treatment effect due to the variation in outcome measures and units of measurement for each outcome among the included studies. For circadian rhythm studies, mean difference was used due to unavailability of necessary data for standardized mean difference. Random effects model was used for meta-analysis with Inverse variance approach. Chi-square and I2 statistics were used as measures of heterogeneity. The DerSimonian and Laird variance estimator and Wald type method was used to calculate the confidence interval for the summary effect. The cut off values for heterogeneity (0–40%: may not be important, 30–60%: may represent moderate heterogeneity, 50–90%:may represent substantial heterogeneity, 75–100%: considerable heterogeneity) were considered as per guidelines mentioned in the Cochrane Handbook for Systematic Reviews of Interventions. Output of meta-analysis is depicted in forest plots, where the diamond shows the pooled estimate with 95% confidence interval. Meta-analysis was performed in Review Manager 5.4.1 software (Cochrane Training 2023).

Results

Study selection

A total of 3,366 articles were retrieved; out of which duplicates were removed, and 3,216 articles were screened for eligibility based on title and abstract. A total of 47 articles were included for full text screening and 11 articles were excluded based on the study design, or interventions other than exercise. Post screening, 36 articles were included for qualitative synthesis and 26 articles, for quantitative synthesis via meta-analysis. No new articles were included when the search was rerun on 27th of July, 2023. The detailed PRISMA flowchart of the screening and selection process has been denoted in Fig. 1.

Figure 1 PRISMA flowchart summarizing the article selection process of the review.

Risk of bias

The risk of bias of the included aerobic exercise, resistance exercise, combined exercise, physical activity, tai chi and yoga studies has been denoted in Supplemental Information S1A–S1F. Of the twelve studies that evaluated the effect of aerobic exercise, high risk of bias was reported for blinding of participants and personnel (Chen et al., 2016; Dodd et al., 2010), incomplete outcome data (Wang et al., 2011) and other bias (Mercier, Ivers & Savard, 2018; Payne et al., 2008). An unclear risk of bias was reported for random sequence generation (Wang et al., 2011; Payne et al., 2008; Cho et al., 2012; Khoirunnisa et al., 2019; Naraphong et al., 2015; Roveda et al., 2017; Wenzel et al., 2013), allocation concealment (Chen et al., 2016; Dodd et al., 2010; Wang et al., 2011; Payne et al., 2008; Cho et al., 2012; Khoirunnisa et al., 2019; Naraphong et al., 2015; Roveda et al., 2017; Wenzel et al., 2013; Tang, Liou & Lin, 2010) and blinding of participants and personnel (Wang et al., 2011; Mercier, Ivers & Savard, 2018; Payne et al., 2008; Cho et al., 2012; Khoirunnisa et al., 2019; Roveda et al., 2017; Wenzel et al., 2013; Tang, Liou & Lin, 2010; Courneya et al., 2012). The resistance exercise study reported unknown risk of bias in random sequence generation, allocation concealment and blinding of participants and personnel (Steindorf et al., 2017).

Of the six studies that evaluated the effect of combined exercise program, an unclear risk of bias was observed for allocation concealment (Coleman et al., 2012; Sprod et al., 2010). For blinding of participants and personnel, a high risk of bias was reported in two (Cheville et al., 2013; Kampshoff et al., 2015) and unclear risk of bias was reported in four studies (Coleman et al., 2012; Sprod et al., 2010; Courneya et al., 2014; Rogers et al., 2015). Among the five physical activity studies, two studies reported high risk of bias for blinding of participants and personnel (Donnelly et al., 2011; Li et al., 2022), while the other three reported an unclear risk of bias (Rogers et al., 2009; Rogers et al., 2013; Rogers et al., 2017). However, it must be acknowledged that blinding of participants is not always possible in exercise trials. Of the four tai chi studies (Irwin et al., 2017; Larkey et al., 2015; McQuade et al., 2017; Lu et al., 2019), one reported high risk of bias for other bias (Larkey et al., 2015) and two reported unclear risk of bias for allocation concealment (Larkey et al., 2015; McQuade et al., 2017). Of the eight studies that evaluated the effect of yoga, high risk of bias was reported for blinding of participants and personnel in three studies (Chandwani et al., 2014; Chaoul et al., 2018; Huberty et al., 2019) and other bias in two studies (Cramer et al., 2016; Taylor et al., 2018). An unclear bias was reported for random sequence generation in two studies (Huberty et al., 2019; Taylor et al., 2018), allocation concealment in three studies (Chandwani et al., 2014; Chaoul et al., 2018; Huberty et al., 2019), blinding of participants and personnel in two studies (Cramer et al., 2016; Vadiraja et al., 2009) and blinding of outcome assessment in one study (Vadiraja et al., 2009).

Population of included studies

Thirty-five of the 36 studies included adult population with age ranging from 18 to 83 years, with one study including a pediatric population aged between 8-18 years (Khoirunnisa et al., 2019). Sixteen studies include breast cancer survivors exclusively (Wang et al., 2011; Payne et al., 2008; Naraphong et al., 2015; Roveda et al., 2017; Steindorf et al., 2017; Courneya et al., 2014; Rogers et al., 2015; Rogers et al., 2009; Rogers et al., 2013; Rogers et al., 2017; Irwin et al., 2017; Larkey et al., 2015; Chandwani et al., 2014; Chaoul et al., 2018; Taylor et al., 2018; Vadiraja et al., 2009), fourteen included mix tumors (Dodd et al., 2010; Mercier, Ivers & Savard, 2018; Cho et al., 2012; Khoirunnisa et al., 2019; Wenzel et al., 2013; Tang, Liou & Lin, 2010; Sprod et al., 2010; Cheville et al., 2013; Kampshoff et al., 2015; Donnelly et al., 2011; Li et al., 2022; McQuade et al., 2017; Cramer et al., 2016; Mustian et al., 2013), two included lymphomas (Courneya et al., 2012; Cohen et al., 2004), with one study each of lung cancer (Chen et al., 2016), myeloploriferative tumors (Huberty et al., 2019), colorectal cancer (Lu et al., 2019) and multiple myeloma (Coleman et al., 2012).

Intervention of included studies

Among the 36 included studies, twelve studied the effect of aerobic exercise (Chen et al., 2016; Dodd et al., 2010; Wang et al., 2011; Mercier, Ivers & Savard, 2018; Payne et al., 2008; Cho et al., 2012; Khoirunnisa et al., 2019; Naraphong et al., 2015; Roveda et al., 2017; Wenzel et al., 2013; Tang, Liou & Lin, 2010; Courneya et al., 2012), one of resistance exercise (Steindorf et al., 2017), six of combined aerobic and resistance exercise (Coleman et al., 2012; Sprod et al., 2010; Cheville et al., 2013; Kampshoff et al., 2015; Courneya et al., 2014; Rogers et al., 2015), five of physical activity intervention (Donnelly et al., 2011; Li et al., 2022; Rogers et al., 2009; Rogers et al., 2013; Rogers et al., 2017), four of tai chi (Irwin et al., 2017; Larkey et al., 2015; McQuade et al., 2017) and eight of yoga (Chandwani et al., 2014; Chaoul et al., 2018; Huberty et al., 2019; Cramer et al., 2016; Taylor et al., 2018; Vadiraja et al., 2009; Mustian et al., 2013; Cohen et al., 2004). The range of duration of the intervention was as follows, aerobic exercise—5 days to 1 year, resistance exercise—12 weeks, combined exercise—3 weeks to 3 months, yoga—4 weeks to 1 year, tai chi—7 weeks to 24 weeks and physical activity—8 weeks to 3 months.

Sleep and circadian rhythm outcomes

A summary of the different sleep outcomes assessed across the different exercise modalities is summarized in Supplemental Information S2A–S2B. Self-reported sleep quality appeared to be the most commonly outcome measures in the studies, with the Pittsburgh Sleep Quality Index (PSQI) by far the most commonly used (Chen et al., 2016; Wang et al., 2011; Mercier, Ivers & Savard, 2018; Payne et al., 2008; Wenzel et al., 2013; Tang, Liou & Lin, 2010; Courneya et al., 2012; Sprod et al., 2010; Kampshoff et al., 2015; Courneya et al., 2014; Rogers et al., 2015; Donnelly et al., 2011; Li et al., 2022; Rogers et al., 2013; Rogers et al., 2017; Irwin et al., 2017; Larkey et al., 2015; McQuade et al., 2017; Lu et al., 2019; Chandwani et al., 2014; Chaoul et al., 2018; Cramer et al., 2016; Mustian et al., 2013; Cohen et al., 2004). Accelerometry was relatively commonly used to provide insight into more objective sleep outcomes, with sleep efficiency (SE) (Chen et al., 2016; Mercier, Ivers & Savard, 2018; Payne et al., 2008; Roveda et al., 2017; Coleman et al., 2012; Rogers et al., 2015; Rogers et al., 2013; Rogers et al., 2017; Irwin et al., 2017; Chaoul et al., 2018) and sleep onset latency (SOL) (Chen et al., 2016; Mercier, Ivers & Savard, 2018; Roveda et al., 2017; Rogers et al., 2015; Rogers et al., 2013; Rogers et al., 2017; Irwin et al., 2017; Chaoul et al., 2018) most commonly reported. Lastly, only five studies assessed any circadian rhythm markers (Chen et al., 2016; Payne et al., 2008; Roveda et al., 2017; Chandwani et al., 2014; Vadiraja et al., 2009). Two studies included accelerometry indicators of circadian rhythm including r24, I < O (Chen et al., 2016) and MESOR (Roveda et al., 2017). Other circadian rhythm markers included serum cortisol, serum serotonin (Payne et al., 2008) and salivary cortisol (Chandwani et al., 2014; Vadiraja et al., 2009).

Effects on sleep and circadian rhythm outcomes

The following sections provide a summary of the effects of the different exercise modes on the sleep and circadian rhythm outcomes. The details of the same have been summarized in Supplemental Information 3–4. Due to the large inter-study variation in exercise modes as well as outcome (sleep and circadian rhythm) measures, it was felt most appropriate to perform meta-analyses on the different outcomes, with sub-analyses presented for the applicable exercise modes for each of the different outcomes.

Effect of exercise on sleep outcomes

SE as assessed by accelerometry

Six studies involving a total of 294 exercise and 307 usual care participants were eligible to be included in the meta-analysis for the effect of exercise on SE, as assessed by accelerometry. The results of these studies indicated no significant effect of exercise on sleep efficiency (SMD = 0.06 (95% CI [−0.17–0.28]), p = 0.61) (Fig. 2).

Figure 2 The effects of various interventions vs usual care on sleep efficiency as assessed by accelerometry in cancer survivors.

Sleep onset latency (SOL) as assessed by accelerometry

Six trials with a total of 294 exercise and 307 usual care individuals were analyzed for the effect of exercise on SOL as measured by accelerometry. Overall, these studies found no significant effect of exercise on SOL (SMD = −0.24 (95% CI [−0.57–0.09]), p = 0.15). However, sub-group analysis revealed that aerobic exercise significantly improved SOL (SMD = −0.53 (95% CI [−0.86 to −0.21]), p = 0.001) and that yoga significantly worsened SOL (SMD = 0.35 (95% CI [0.04–0.66]), p = 0.03) (Fig. 3).

Wake after sleep onset (WASO) assessed by accelerometry

Two studies involving a total of 130 exercise and 140 usual care participants were included for the meta-analysis to analyze the effect of exercise interventions on WASO, as assessed by accelerometry. There was an overall significant reduction in WASO observed with exercise (SMD = −0.29 [−0.53, −0.05], p = 0.02) (Fig. 4).

Figure 3 The effect of various interventions vs usual care on sleep onset latency as assessed by accelerometry in cancer survivors.

Figure 4 The effect of various interventions vs usual care on wake after sleep onset (WASO) as assessed by accelerometry in cancer survivors.

Total sleep time (TST) assessed by accelerometry

TST was analyzed in two studies involving a total of 130 exercise and 140 usual care participants. The meta-analyses showed no overall effect of exercise on TST as assessed by accelerometry (SMD = 0.10 [−0.14, 0.34], p = 0.40) (Fig. 5).

Figure 5 The effect of various interventions vs usual care on total sleep time (TST) as assessed by accelerometry in cancer survivors.

Sleep quality as assessed by PSQI

Twenty-one studies involving a total of 1,138 exercise and 1,127 usual care participants were eligible for meta-analysis for the effect of exercise on sleep quality assessed by PSQI. There was a significant overall effect of exercise for improving sleep quality as reported by PSQI (SMD = −0.50 [−0.87, −0.13], p = 0.008). Subgroup analysis also highlighted the significant improvement seen in sleep quality with combined exercise (SMD = −0.25 [−0.43, −0.06], p = 0.009) and tai chi (SMD = −1.07 [−1.93, −0.21], p = 0.01) (Fig. 6).

Figure 6 The effect of various interventions vs usual care on sleep quality as assessed by PSQI in cancer survivors.

Sleep quality as assessed by General Sleep Disturbance Scale (GSDS)

Two studies including 63 intervention and 77 usual care participants were included in the meta-analysis for their effects on sleep quality as measured by GSDS. Results indicated that these two aerobic studies had no significant effect on sleep quality as assessed by the GSDS (SMD = 0.07 [95% CI −0.26, 0.40], p = 0.68) (Fig. 7).

Figure 7 The effect of yoga vs usual care on circadian rhythm as assessed by salivary cortisol levels in cancer survivors.

Effect of exercise on circadian rhythm outcomes

Five studies including 192 exercise and 194 usual care participants assessed the effect on exercise interventions on circadian rhythm. Two of the studies used accelerometry (Chen et al., 2016; Roveda et al., 2017), two studies used salivary cortisol levels (Chandwani et al., 2014; Vadiraja et al., 2009), with serum cortisol and serum serotonin levels also assessed in one study (Payne et al., 2008), to estimate the regulation of circadian rhythm. Among the five studies, three studied the effect of aerobic exercise (Chen et al., 2016; Payne et al., 2008; Roveda et al., 2017) and two studied the effect of yoga (Chandwani et al., 2014; Vadiraja et al., 2009) on circadian rhythm indicators.

Circadian rhythm as assessed by salivary cortisol values

The effect of yoga on circadian rhythm assessed by salivary cortisol levels in 58 intervention and 57 usual care participants, showed a significant improvement (MD = −0.09 (95% CI [−0.13 to −0.06]) mg/dL, p < 0.001) (Fig. 8).

Figure 8 The effect of aerobic training interventions vs usual care on sleep quality as assessed by GSDS in cancer survivors.

Circadian rhythm as assessed by accelerometry values

Effect of aerobic exercise on circadian rhythm was assessed by accelerometry in two studies with 75 participants in the intervention and 76 in the usual care group. No significant effect of aerobic exercise was observed (MD = 2.23 (95% CI [−3.45–7.91]), p = 0.44) (Fig. 9).

Figure 9 The effect of aerobic training vs usual care on circadian rhythm as assessed by accelerometry values (r24, MESOR) in cancer survivors.

Effect of exercise on other sleep and circadian rhythm outcomes

Ten of the eligible 36 studies were not included in the meta-analysis due to a variety of reasons including unique outcome measures which were not used in any other study, a lack of a usual care control group or lack of statistical data required for meta-analysis despite reaching out to the authors. Eight of the ten studies compared aerobic exercise (Dodd et al., 2010; Payne et al., 2008; Khoirunnisa et al., 2019), resistance exercise (Steindorf et al., 2017), combined exercise (Coleman et al., 2012; Cheville et al., 2013) or yoga (Huberty et al., 2019; Taylor et al., 2018) to usual care, while two studies compared aerobic exercise (Mercier, Ivers & Savard, 2018) or tai chi (Irwin et al., 2017) to CBT.

Studies with usual care as comparison

Two of the eight studies that compared the effect of aerobic exercise with usual care (Dodd et al., 2010; Payne et al., 2008) on sleep quality as assessed by PSQI were excluded from the meta-analysis as insufficient data was provided for meta-analysis. One of these studies demonstrated a statistically significant improvement in sleep quality (Payne et al., 2008). Two of the eight studies studied the effect of aerobic exercise (Payne et al., 2008) or combined exercise (Coleman et al., 2012) on a variety of sleep outcomes as assessed by accelerometry. One of these studies showed statistically significant improvement in actual wake time and movement in sleep (Payne et al., 2008). Sleep quality assessed by 11 point NRS in adults (Cheville et al., 2013) and SDS for children (Khoirunnisa et al., 2019), showed significant improvement in combined exercise or aerobic exercise group, respectively when compared to usual care. In contrast, sleep quality assessed by Sleep disturbance short form 8A (Huberty et al., 2019) and insomnia severity assessed by Insomnia Severity Index (Mercier, Ivers & Savard, 2018) failed to show statistical significance when yoga was compared to usual care. Resistance exercise was prescribed only in one study and showed statistically significant benefits in sleep quality as assessed by European Organization for the Research and Treatment of Cancer Quality of Life Questionnaire (EORTCQLQ 30) Insomnia subscale (Steindorf et al., 2017).

Circadian rhythm as assessed by serum cortisol and serum serotonin was assessed in one study, whereby only serum serotonin showed a statistically significant decrease (Payne et al., 2008).

Studies with CBT as comparison

Both the studies that evaluated the effect of yoga or tai chi on sleep quality, as assessed by subjective and objective outcome measures, demonstrated that aerobic exercise or tai chi were not inferior to CBT-1 in improving these sleep outcomes (Mercier, Ivers & Savard, 2018; Irwin et al., 2017).

Discussion

Sleep disturbance and circadian rhythm de-regulation are common symptoms that negatively affect recovery and mortality in many groups of cancer survivors during and after treatment. We feel that this systematic review adds substantially to the literature as it includes many forms of exercise and a wide variety of sleep disturbance and circadian rhythm outcomes in cancer survivors. Of the 36 studies included in this review, 26 were included in the meta-analyses. Meta-analyses were performed exclusively for the most common sleep and circadian rhythm outcomes across all exercise modes, with some sub-analyses also performed for the individual exercise modes. Such an approach was performed to identify which: (1) sleep and circadian rhythm outcomes may be improved by exercise in cancer survivors; and (2) exercise mode may be most effective for improving each of the different outcomes, respectively.

The meta-analysis in this review showed an overall significant beneficial effect of exercise on sleep quality and circadian rhythm, as assessed by the PSQI, WASO and salivary cortisol levels, respectively. The potentially most clinically relevant of these benefits was the significant, moderate exercise-related improvements in self-reported sleep quality, assessed by the PSQI. These clinical benefits may not only reflect the improvement in many direct and indirect adverse events associated with poor sleep quality in cancer survivors such as reduced appetite, weaker immunity, risk of high blood pressure, anxiety and depression (National Cancer Institute, 2015), but also result in better prognosis and survival among cancer survivors (Collins et al., 2017). Further, these findings are in line with results for mixed population (older adults with metabolic syndrome, depression, prostate cancer, knee osteoarthritis, fibromyalgia, post-stroke, or breast cancer), where sleep quality, as assessed by PSQI was significantly improved by combined exercise or tai chi interventions (National Cancer Institute, 2015; Collins et al., 2017; Zhou et al., 2022; Li et al., 2020).

In contrast, our meta-analysis indicated no significant improvements for SE, SOL, TST, sleep quality as assessed by the GSDS and accelerometry-derived circadian rhythm outcomes. The relative equivalence of our meta-analysis (and that of the wider exercise sleep literature), whereby some sleep and circadian outcomes were significantly improved but others showed no significant exercise-related change, may reflect a number of methodological issues within the studies. One such issue could be the wide variety of mechanisms contributing to sleep and circadian rhythm disturbances and how the relative influence of these mechanisms likely differs across various medical conditions including cancer and their common treatments (Balachandran et al., 2021). As a result, Balachandran et al. (2021) have developed a clinical algorithm for the evaluation of sleep disorders in cancer survivors, potentially involving surveys, physical exams, medications, imaging, and pulmonary, electrocardiography and laboratory studies, which can be used to better identify the major issues and inform the appropriate treatment plan for each cancer survivor. However, the exercise trials that were included in our and other meta-analysis typically applied a general exercise program to all of the participants who meet some broad inclusion and exclusion guidelines, without performing any assessments recommended by Balachandran et al. (2021) to identify their major sleep disturbances and the mechanisms contributing to these sleep-related symptoms. A second issue was the relatively large between-study variation in participant characteristics (e.g., cancer type, age, previous and current treatments, severity of sleep and circadian rhythm disturbances), exercise characteristics (mode, duration, frequency and intensity), sleep outcomes as well as assessments for each sleep outcome. In terms of cancer type, a recent meta-analysis of six studies examined whether physical activity could improve sleep outcomes in breast cancer survivors (Yang et al., 2021). While significant physical activity related improvements in sleep quality as assessed by the PSQI were observed, no significant benefits for a variety of other sleep outcomes including global PSQI score, sleep duration, sleep medication, sleep latency, habitual sleep efficiency and daytime dysfunction. Unfortunately, relatively substantial interstudy variation was found in the exercise prescription for these studies involving breast cancer survivors, with mixtures of supervised and home-based exercise, resistance training, aerobics exercise and general physical activity. Collectively, these two issues make a strong case for whereby cancer exercise trials wishing to demonstrate significant benefits for improved sleep and circadian rhythm outcomes may need to have stricter eligibility criteria (e.g., a minimum level of sleep and circadian rhythm disturbance at baseline) and then match the most appropriate forms of exercise to the primary mechanisms contributing to these particular disturbances in each cancer survivor (or homogenous group of cancer survivors).

The other major issue may then reflect the wide variety of outcomes used to quantify sleep and circadian rhythm disturbances and the actual assessments for each of these outcomes. For example, while clinicians typically prefer to base their treatment plans on objective data (such as polysomnography for sleep disorders), self-reported data from the cancer survivor is also highly important as it provides a measure of how they currently feel. One issue with the objective measures typically used in these studies, is the use of accelerometry rather than polysomnography, which is considered the criterion measures for insomnia research and clinical practice (Douglas et al., 2017).

Strengths and limitations

As this systematic review and meta-analysis includes studies involving a variety of forms of exercise and cancer survivors with a variety of ages, cancer stage and treatment status, this systematic review has attempted to serve as a reference for prescription of exercise for cancer survivors during and after their cancer treatment. However, the limitations of this inclusive approach includes heterogeneity regarding the type of cancer, baseline sleep and circadian rhythm disturbance characteristics of the participants, exercise intervention as well as sleep outcomes and assessment tools in the included studies. It should however be acknowledged that the heterogeneity of exercise mode is partially offset for each sleep and circadian rhythm outcome, as these meta-analyses involved exercise-mode specific, sub-group analyses for each different exercise mode. Future research needs to look at particular forms of exercise and specific cancer groups to better determine what types of exercise may improve the wide variety of sleep and circadian rhythm outcomes. These studies may also need to better define what level of sleep or circadian rhythm disturbance participants need to demonstrate to be eligible for the studies, as these baseline sleep and circadian rhythm characteristics could be important cofounders that need to be more clearly controlled to better understand the effect of various exercise prescriptions. It should also be acknowledged that we did not utilise the Cochrane Central Register of Controlled Trials (CENTRAL) databases in our search, meaning that some eligible studies may have been missed. This was due to the unavailability of access to CENTRAL in few countries where the reviewers were located (India). However, since there was considerable overlap in the databases we utilised and what is found in CENTRAL, it is hoped no eligible studies were missed.

Conclusions

Based on the meta-analysis results, sleep quality and circadian rhythms as assessed by PSQI, WASO and salivary cortisol levels, respectively have shown to significantly improve with exercise interventions in cancer survivors. However, such overall exercise effects tended to be only observed with a select number of sleep and circadian rhythm outcomes as well as exercise modes. Based on the wide range of benefits of exercise and physical activity for cancer survivorship, it may still be worthwhile for cancer survivors with sleep and circadian rhythm disturbances to be encouraged to regularly perform the form of exercise or regular physical activity they prefer in their weekly activities.

Supplemental Information

Supplemental Information 1 Raw Data

Supplemental Information 2 Risk of bias of aerobic exercise studies

Supplemental Information 3 Risk of bias of resistance exercise studies

Supplemental Information 4 Risk of bias of combined exercise studies

Supplemental Information 5 Risk of bias of physical activity studies

Supplemental Information 6 Risk of bias of tai chi studies

Supplemental Information 7 Risk of bias of yoga studies

Supplemental Information 8 Summary of outcome measures used for sleep quality

Supplemental Information 9 Summary of outcome measures used for circadian rhythm

Supplemental Information 10 Summary of the methods and intervention for the exercise studies

Supplemental Information 11 Summary of results of the exercise studies

Supplemental Information 12 PRISMA checklist

Supplemental Information 13 Systematic Review and/or Meta-Analysis Rationale

Additional Information and Declarations

Competing Interests

Author Contributions

Data Availability

Justin Keogh is an Academic Editor for PeerJ.

Rachita Gururaj conceived and designed the experiments, performed the experiments, prepared figures and/or tables, authored or reviewed drafts of the article, and approved the final draft.

Stephen Rajan Samuel conceived and designed the experiments, performed the experiments, prepared figures and/or tables, authored or reviewed drafts of the article, and approved the final draft.

K Vijaya Kumar performed the experiments, prepared figures and/or tables, and approved the final draft.

Ravishankar Nagaraja performed the experiments, analyzed the data, prepared figures and/or tables, and approved the final draft.

Justin W.L. Keogh conceived and designed the experiments, performed the experiments, prepared figures and/or tables, authored or reviewed drafts of the article, and approved the final draft.

The following information was supplied regarding data availability:

The raw measurements are available in the Supplementary File.

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
