# Peer review of "Effect of exercise based interventions on sleep and circadian rhythm in cancer survivors—a systematic review and meta-analysis"

_PeerJ, doi:10.7717/peerj.17053_

## Round 0.1 · original submission · Major Revisions

The reviewers found merit in the manuscript. However, some major issues were pointed out and should be addressed by the authors. Please, be careful about the clarification of methods and procedures, that makes the manuscript valid, reliable, and reproducible. More details should be provided on this. Remember the paper can still be rejected if the reviewers are not impressed with the sophistication of the revisions made.

·

Basic reporting

--the detailed search strategy should be organized as a supplementary file.
--In the "Materials & Methods" section, please add outcome variables.

Experimental design

no comment

Validity of the findings

no comment

Additional comments

--In the discussion section, it is necessary to add an analysis of the impact of exercise on sleep quality for specific types of cancer, especially the related meta-analysis articles.
eg.DOI: 10.1007/s00520-020-05914-y
https://doi.org/10.1007/s10552-019-01156-4

**Staff Note: Regarding these citation suggestions, it is PeerJ policy that additional references suggested during the peer-review process should only be included if the authors are in agreement that they are relevant and useful**

Reviewer 2 ·

Basic reporting

I believe a definition of sleep disturbance in line 69 is warranted.

In line 62, please clarify whether you mean that cancer survivors are more likely to present with disrupted circadian rhythms than the general population.

Experimental design

Please include more information about how many additional articles were included after the literature search was re-run in 2023.

Validity of the findings

No comment

Additional comments

Please clarify why the study including a pediatric population (line 218) was included.

·

Basic reporting

General recommendation: please re-check each PRISMA item and report accordingly. In some items the page numbers noted do not provide the full required information.

The specific comments below are just examples and not exhaustive:

Please check and rewrite (if necessary) the first sentence in the abstract. Isn't there a with too many? "Disrupted circadian rhythm is commonly reported in cancer survivors, with this closely associated with cancer related fatigue, sleep disturbances and compromised quality of life"

Line 68/69: ... are believed to [be??]

Item 7 of the PRISMA checklist http://prisma-statement.org/documents/PRISMA_2020_expanded_checklist.pdf

Headers: please consider to write out words in headers (for exemple line 255 SOL).
Line 110: Please provide the full search strategy (including the database-specific field desdcriptors) for all databases, as required in the PRISMA checklist.

Line 133. Please, be more specific about quasi-RCT design. what would you consider a quasi-RCT design and what do you consider a true RCT design.
Line 14ff: Please provide more detailled information regarding data extraction, eg. (but not exlusive, see PRISMA checklist) on how the two people extracted data (double data extraction, one extraction, one controlled/checked, or both extracted some data so that each extractata data was only extracted from one author).
Line 147: how did you decide on which outcome-tool and which time-point to extract if there where more than one outcome-measurement per outcome-construct and more than one time-point (hierarchy of outcome-measures and time-points)?

Line 169ff: could you please give more information according to PRISMA item 13d on statistical synthesis method (see e.g. http://prisma-statement.org/documents/PRISMA_2020_expanded_checklist.pdf)

Risk of bias analysis: Please reconsider the rating of the masking of participants item; how is it possible to mask participants in exercise trials? Why ? in many studies for this item.
Risk of bias analysis: Please reconsider the rating of observer masking. If the patient is not blinded and the outcome is a self-reported questionnaire, how can the observe be masked (here the patient would be the observer).

Experimental design

Method section: One limitation is that the authors did not search Cochrane Central Register of Controlled Trials (CENTRAL).
Limitation data extraction: Not sure whether data extraction was done in double or at least controlled by someone.

Validity of the findings

Line 360 / 361 Could you please make sure that the reader does not mistake this statement..."but also result in better
361 prognosis and survival among cancer survivors[58]" -->how strong is this evidence from reference 58.

---

## Round 0.2 · accepted · Accept

The authors addressed all the reviewers' comments. Congratulations on the improved manuscript. Some minor changes can be made during the following steps (e.g., correction of Figure 1 caption).

·

Basic reporting

no comment

Experimental design

no comment

Validity of the findings

no comment

Additional comments

no comment

Reviewer 2 ·

Basic reporting

No comment

Experimental design

No comment

Validity of the findings

No comment

Additional comments

N/A

·

Basic reporting

Thank you very much for the changes made.
The only remaining problem from my point of view is the rating of the risk of bias, specially the blinding of the assessors. For the subjective sleep outcomes, the assesor is the patient. And if he or she is not blinded, the assessor will not be blinded. But i accept that this might be seen as a matter of opinition.

Experimental design

ok

Validity of the findings

ok

Additional comments

nothing